# Expression of SREBP-1c Requires SREBP-2-mediated Generation of a Sterol Ligand for LXR in Livers of Mice

**Shunxing Rong[1], Víctor A Cortés[1†], Shirya Rashid[1‡], Norma N Anderson[1], Jeffrey G McDonald[1], Guosheng Liang[1], Young-Ah Moon[1§], Robert E Hammer[2], Jay D Horton[1,3]***

[1]Department of Molecular Genetics, University of Texas Southwestern Medical Center, Dallas, United States; [2]Department of Biochemistry, University of Texas Southwestern Medical Center, Dallas, United States; [3]Department of Internal Medicine, University of Texas Southwestern Medical Center, Dallas, United States

**\*For correspondence:** Jay. Horton@utsouthwestern.edu

**Present address:** [†]Department of Nutrition, Diabetes and Metabolism, School of Medicine, Pontificia Universidad Católica de Chile, Santiago, Chile; [‡]Department of Public Health, North South University, Bashundhara, Dhaka, Bangladesh; [§]Department of Biomedical Sciences, Inha University College of Medicine, Nam-gu Iuncheon, Korea

**Competing interests:** The authors declare that no competing interests exist.

**Abstract** The synthesis of cholesterol and fatty acids (FA) in the liver is independently regulated by SREBP-2 and SREBP-1c, respectively. Here, we genetically deleted *Srebf-2* from hepatocytes and confirmed that SREBP-2 regulates all genes involved in cholesterol biosynthesis, the LDL receptor, and PCSK9; a secreted protein that degrades LDL receptors in the liver. Surprisingly, we found that elimination of *Srebf-2* in hepatocytes of mice also markedly reduced SREBP-1c and the expression of all genes involved in FA and triglyceride synthesis that are normally regulated by SREBP-1c. The nuclear receptor LXR is necessary for *Srebf-1c* transcription. The deletion of *Srebf-2* and subsequent lower sterol synthesis in hepatocytes eliminated the production of an endogenous sterol ligand required for LXR activity and SREBP-1c expression. These studies demonstrate that cholesterol and FA synthesis in hepatocytes are coupled and that flux through the cholesterol biosynthetic pathway is required for the maximal SREBP-1c expression and high rates of FA synthesis.

## Introduction

Cholesterol and fatty acid (FA) biosynthetic gene expression is regulated by the sterol regulatory element-binding protein (SREBP) family of transcription factors (*Horton et al., 2002*). The three family members—SREBP-1a, SREBP-1c, and SREBP-2—are basic-helix-loop helix transcription factors that bind to sterol response elements of promoters to activate transcription. SREBP-1a and SREBP-1c are encoded by the same gene but have independent promoters that utilize a unique first exon. SREBP-2 is encoded by a separate gene.

The membrane-bound, inactive forms of SREBPs are located in the endoplasmic reticulum bound to Scap, an escort protein that serves as a sensor of cellular sterol levels (*Brown and Goldstein, 2009*). When cellular sterol levels are high, Scap binds an ER retention protein, Insig, which retains the SREBP/Scap complex in the ER. To generate the active nuclear form of SREBPs, SREBP/Scap dissociates from Insig, and the complex moves from the ER to the Golgi where two proteases, designated S1P and S2P, sequentially cleave SREBPs releasing the amino-terminal fragment, which travels to the nucleus to activate regulated genes.

The in vivo transcriptional-activating properties of each SREBP isoform have been investigated through the generation and characterization of transgenic and knockout mice (*Horton et al., 2002*). In most tissues, the predominant SREBP-1 isoform expressed is SREBP-1c (*Shimomura et al., 1997*). Overexpression of nuclear SREBP-1c (nSREBP-1c) in livers of mice resulted in the transcriptional

activation of genes involved in FA and triglyceride (TG) synthesis (*Horton et al., 2002*). As a consequence of increased de novo lipogenesis, mice expressing nSREBP-1c developed fatty livers. Consistent with the role of SREBP-1c in activating lipogenesis, SREBP-1c is activated in the liver by insulin through a partially defined pathway that involves the insulin receptor, Akt, and mTORC1 (*Owen et al., 2012*). Conversely, genetic deletion of *Srebf-1c* led to a selective reduction in the expression of genes involved in FA and TG synthesis (*Liang et al., 2002*).

The SREBP-1a isoform is a more potent transcription activator than SREBP-1c, owing to its longer transactivation domain (*Horton et al., 2002*). Overexpression of the minor nSREBP-1a isoform in mouse liver led to the activation of genes involved in both FA and cholesterol biosynthesis; resulting in the accumulation of both cholesterol and TGs in liver (*Horton et al., 2002*). In stark contrast, the selective deletion of *Srebf-1a* reduced the expression of only acetyl-CoA carboxylase (ACC) 2 in the liver, one of the two ACC isoforms that carry out the first committed enzymatic step in FA synthesis (*Im et al., 2009*).

The genetic ablation of both *Srebf-1a* and *Srebf-1c* resulted in significant, but incomplete, embryonic lethality (*Horton et al., 2002*). In those few *Srebf-1a/Srebf-1c* knockout mice that survived to adulthood, the gene expression profile in the liver was similar to that observed in livers from mice that had the genetic ablation of only the *Srebf-1c* isoform.

Transgenic overexpression of nSREBP-2 in liver led to the preferential activation of genes involved in cholesterol biosynthesis, the LDL receptor (LDLR), and PCSK9 (*Horton et al., 2003*). However, nSREBP-2 overexpression also increased the mRNA levels of FA biosynthetic genes, albeit to a lesser extent than those involved in cholesterol synthesis.

To further delineate genes specifically regulated by SREBP-2, we initially attempted to obtain mice with homozygous germ-line deletions of *Srebf-2* using a traditional gene-replacement approach. Crosses of *Srebf-2*[+/-] mice did not produce viable offspring homozygous for the disrupted *Srebf-2* allele. Most embryos homozygous for the disrupted *Srebf-2* allele appeared to die between day 7–8 post-coitum, but the cause of this embryonic lethality was not investigated (*Horton et al., 2002*). Mice that were heterozygous for the germ-line deletion of *Srebf-2* had no discernable phenotype.

To bypass the embryonic lethality, here we used albumin-driven, Cre-mediated recombination to delete *Srebf-2* in hepatocytes of mice. The results confirm that SREBP2 is required for normal levels of cholesterol biosynthetic gene expression, but unexpectedly, we found the expression of *Srebf-1c* and its target genes for FA and TG synthesis was also dependent on SREBP-2 expression. The absence of SREBP-2 lead to reduced LXR activity, which explained the loss of SREBP-1c expression, possibly owing to the loss of an endogenous sterol ligand that is dependent on flux through the cholesterol biosynthesis pathway.

## Results

The vector and targeting strategy used to conditionally disrupt *Srebf-2* is shown in *Figure 1A and B*. Mice homozygous for the floxed *Srebf-2* allele, were bred to transgenic mice that express Cre recombinase driven by the albumin promoter to obtain hepatocyte-specific gene deletion (hepatocyte-*Srebf-2*[-/-]). Littermates bearing two floxed *Srebf-2* alleles with no albumin-cre were designated as wild type controls. Hepatocyte -*Srebf-2*[-/-] mice weighed slightly less than littermate controls but liver weights were unchanged (*Table 1*). In the absence of SREBP-2, plasma and liver cholesterol concentrations were reduced by 68% and 20%, respectively. Unexpectedly, plasma and liver TGs were also reduced by 50% and 76%, in hepatocyte -*Srebf-2*[-/-] mice (*Table 1*).

Immunoblot analyses of SREBPs from livers of mice described in *Table 1* are shown in *Figure 1C*. As expected, the precursor (P) and nSREBP-2 (N) protein were undetectable in hepatocyte-*Srebf-2*[-/-] livers. However, the SREBP-1 precursor and nuclear protein levels were also reduced by ~90% in hepatocyte-*Srebf-2*[-/-] livers. Calnexin and CREB were used as controls for membrane and nuclear proteins, respectively.

*Figure 2* shows the results of quantitative PCR assays that measured mRNA levels of lipid metabolism related genes in livers of mice described in *Table 1*. The mRNA levels of SREBP-2-regulated genes involved in cholesterol biosynthesis and uptake (HMG-CoA synthase, HMG-CoA reductase, farnesyl diphosphate synthase, squalene synthase, and PCSK9) were reduced by 60–80% in hepatocyte-*Srebf-2*[-/-] livers compared to controls. The mRNA for the LDLR was only reduced by 20%.

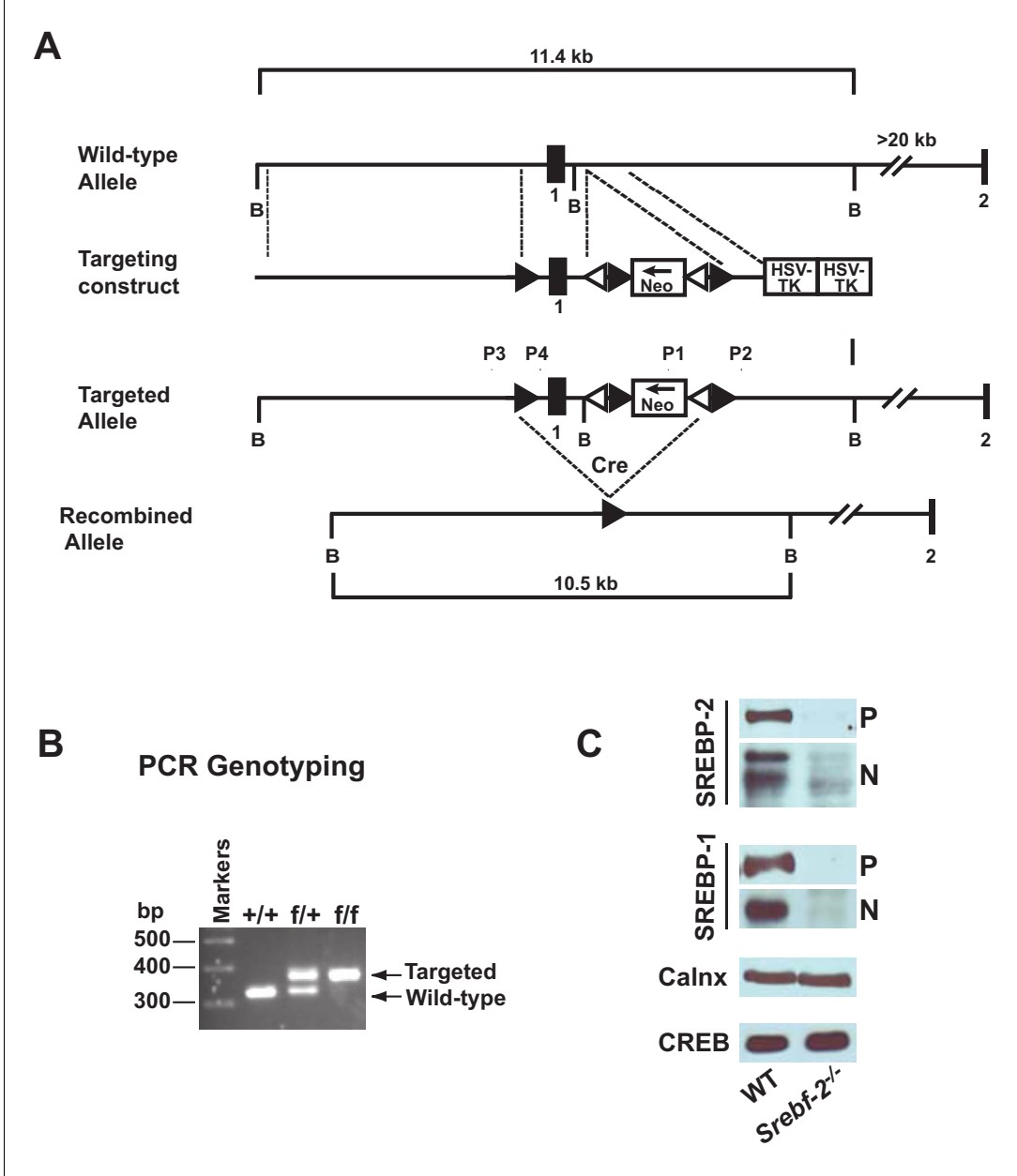

**Figure 1.** Gene-targeting strategy and characterization of the floxed *Srebf-2* allele. (**A**) Schematic of gene-targeting strategy. Cre-mediated excision of the sequences flanked by the loxP sites deletes 660 bp of the *Srebf-2* promoter and exon 1, which includes the initiator methionine and residues encoding the NH₂-terminal domain of *Srebf-2*. The positions of primers (P1 and P2, P3 and P4) used for PCR detection of homologous recombination are denoted by arrowheads. (**B**) Genotype analysis of the conditionally targeted *Srebf-2* mice by PCR of tail-derived DNA. (**C**) Levels of proteins in the livers of WT and hepatocyte-*Srebf-2⁻/⁻* mice. Nuclear and membrane protein was made from each mouse liver described in *Table 1* and equal aliquots from each were pooled (total, 30 μg) and subjected to SDS-PAGE and immunoblot analysis was carried out for the indicated protein as described in 'Materials and methods.' The precursor and nuclear form of SREBPs were denoted as P and N, respectively.

SREBP-1c mRNA levels also were 90% lower than that measured in livers of wild type (WT) mice, while SREBP-1a mRNA levels remained unchanged. SREBP-1c-regulated genes in the FA biosynthetic pathway (ACC1), fatty acid synthase (FAS), ELOVL6, and stearoyl-CoA desaturase-1 (SCD1)) were reduced ~50 to>95% in hepatocyte-*Srebf-2⁻/⁻* livers; however, ACC2 expression, which is primarily regulated by SREBP-1a (*Im et al., 2009*), was only slightly lower. *Srebf-1c* transcription is regulated by LXR and by nSREBP-1c itself through a feed-forward loop (*Repa et al., 2000*). SREBP-1c

**Table 1.** Phenotypic comparison of WT and hepatocyte-*Srebf-2*$^{-/-}$ mice. Male mice 12–13 wks of age fed chow *ad lib* were sacrificed and blood and tissues obtained. Each value represents mean ± SEM.

| Parameters | WT | *Srebf-2*$^{-/-}$ |
|---|---|---|
| Number of mice | 6 | 6 |
| Body weight (g) | 33.1 ± 1.0 | 27.7 ± 1.0* |
| Liver weight (g) | 1.32 ± 0.13 | 1.28 ± 0.09 |
| Plasma cholesterol (mg/dl) | 104 ± 12.3 | 33.7 ± 6.6* |
| Plasma TGs (mg/dl) | 94.8 ± 12.5 | 47.7 ± 1.4* |
| Liver cholesterol (mg/g) | 2.21 ± 0.08 | 1.78 ± 0.06* |
| Liver TGs (mg/g) | 12.4 ± 3.09 | 2.98 ± 0.72* |

*Denotes the level of statistical significance of $p < 0.05$ (Student's *t* test) between WT and hepatocyte-*Srebf-2*$^{-/-}$ mice.

mRNA levels were reduced by 90%, which explains the loss of SREBP-1 protein in hepatocyte-*Srebf-2*$^{-/-}$ livers. The mRNA levels of LXRα and β were unchanged but the mRNA levels of additional LXR-regulated genes, ABCG5 and ABCG8, were reduced by 60–70% (*Supplementary file 2*), suggesting that LXR activity was lower in the absence of SREBP-2.

To confirm that the reduced expression of cholesterol and FA synthesis genes in hepatocyte-*Srebf-2*$^{-/-}$ livers translated into lower rates of lipid synthesis, we measured the incorporation of tritiated water into newly synthesized sterols and FAs. In hepatocyte-*Srebf-2*$^{-/-}$ livers, rates of sterol and FA synthesis were decreased by 59% and 68%, respectively (*Figure 3*).

Inasmuch as the expression of LXRα and β were unaffected by deleting *Srebf-2*, we hypothesized that the loss of SREBP-1c expression and reduced FA synthesis in hepatocyte-*Srebf-2*$^{-/-}$ livers was due to the absence of a ligand for LXR that is either generated within or derived from the cholesterol biosynthetic pathway. To test this hypothesis, we first fed mice a synthetic ligand for LXR, *T0901317*. Administration of *T0901317* to hepatocyte-*Srebf-2*$^{-/-}$ mice induced SREBP-1c mRNA and protein expression to levels similar to that measured in WT livers (*Figure 4A,B*). Increased SREBP-1c expression was associated with higher mRNA levels of ACC1 and FAS (*Figure 4B*). Inasmuch as LXR can independently transcriptionally activate the same FA synthesis genes, we verified that the induction of ACC1 and FAS was specifically due to SREBP-1c by feeding mice that lack all SREBPs as a result of the deletion of *Scap* the LXR agonist (*Moon et al., 2012*). Administration of *T0901317* to mice with hepatocyte-specific deletion of *Scap* did not significantly change the mRNA levels of ACC1 or FAS (*Figure 4—figure supplement 1*). This suggests that LXR administration to the hepatocyte-*Srebf-2*$^{-/-}$ mice induced the mRNA levels of FA synthesis genes through the restoration of SREBP-1c expression and not through direct transcriptional activation by LXR.

Cholesterol feeding leads to the production of oxysterols in the liver that can also activate LXR; therefore, we next fed mice diets supplemented with cholesterol to determine whether dietary cholesterol could restore SREBP-1c expression in hepatocyte-*Srebf-2*$^{-/-}$ livers. Dietary supplementation of 0.2% cholesterol increased liver cholesterol concentrations and SREBP-1c mRNA levels to that measured in WT mice fed chow (*Figure 5A,C*). As shown in *Figure 5B*, nSREBP-1c protein levels in hepatocyte-*Srebf-2*$^{-/-}$ livers were slightly lower than that in WT mice fed chow, but this was sufficient to restore the expression of mRNAs for FA biosynthetic genes to levels found in WT livers (*Figure 5C*). SREBP-2 regulated genes remained low and unaffected by cholesterol feeding (*Figure 5C*).

To identify the potential missing LXR ligand in hepatocyte-*Srebf-2*$^{-/-}$ mice, we performed LC-MS/MS to quantify the cholesterol biosynthetic intermediates and oxysterol concentrations in the liver. As shown in *Supplementary file 1*, the concentrations of intermediates in the cholesterol biosynthetic pathway were not consistently changed or slightly *higher* in livers of hepatocyte-*Srebf-2*$^{-/-}$ mice compared to controls. The cholesterol biosynthetic intermediate, desmosterol, has been previously identified as an LXR ligand (*Yang et al., 2006*); however, the concentration of this intermediate

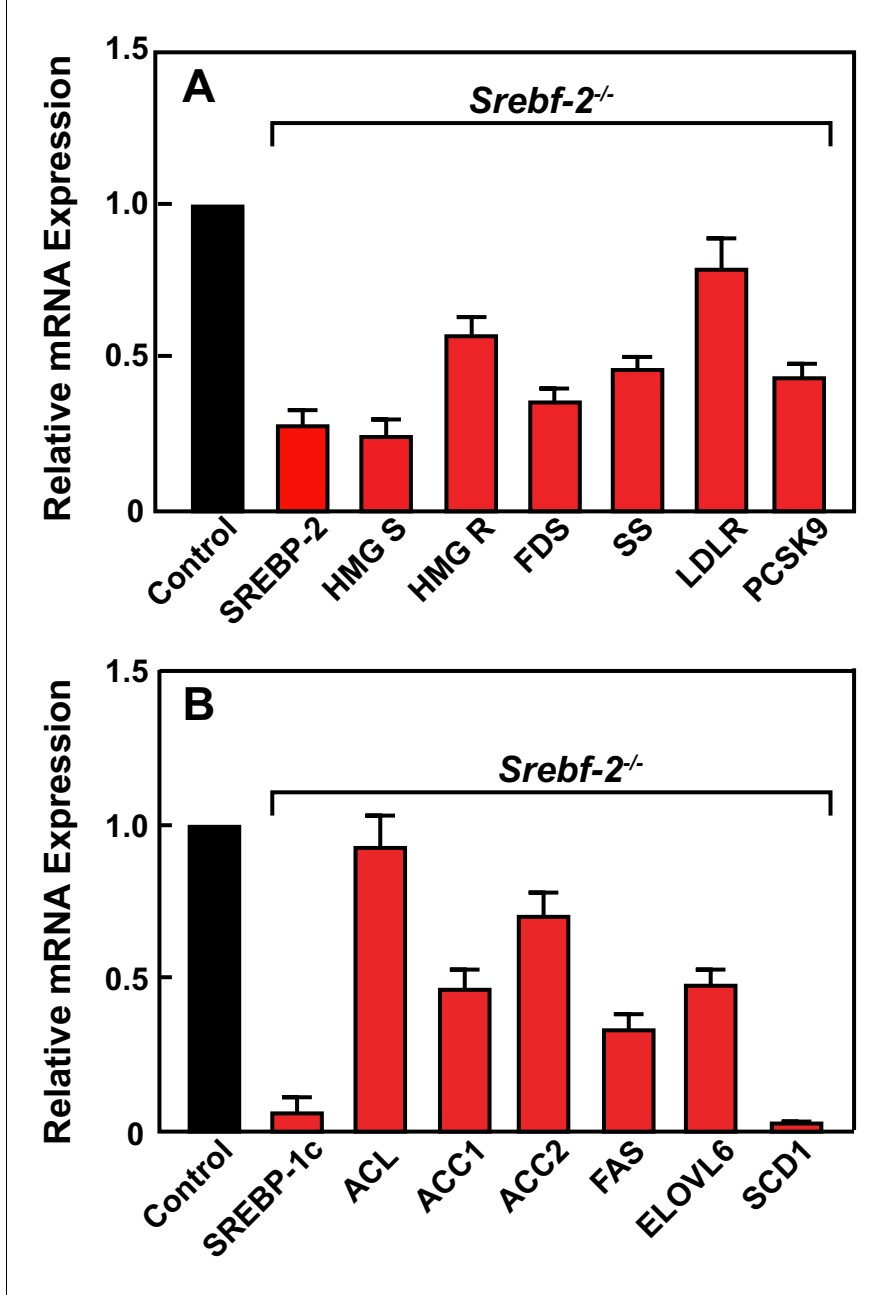

**Figure 2.** Levels of mRNAs in livers of WT and hepatocyte-*Srebf-2*[-/-] mice. Total RNA from livers of each mouse liver described in *Table 1* was subjected to real-time RT-PCR as described in 'Materials and methods.' Apo B was used as the invariant control. Values represent the amount of mRNA relative to those in the wild-type mice, which are arbitrarily assigned a value of 1. (**A**) Genes involved in cholesterol homeostasis. (**B**) Genes involved in FA homeostasis.

was actually higher in hepatocyte-*Srebf-2*[-/-] livers. Other reported ligands of LXR include: 20(S)-hydroxycholesterol, 22(R)-hydroxycholesterol, 24(S)-hydroxycholesterol, 24(S),25-epoxycholesterol, 25-hydroxycholesterol, and 27-hydroxycholesterol (*Huang, 2014*; *Yang et al., 2006*). Of these ligands, 20(S)-hydroxycholesterol and 22(R)-hydroxycholesterol were not detected and concentrations of 24(S)-hydroxycholesterol, 24(S),25-epoxycholesterol, 25-hydroxycholesterol, and 27-hydroxycholesterol. 20(S)-hydroxycholesterol and 22(R)-hydroxycholesterol were either not consistently changed or slightly higher in hepatocyte-*Srebf-2*[-/-] livers compared to controls, suggesting that the

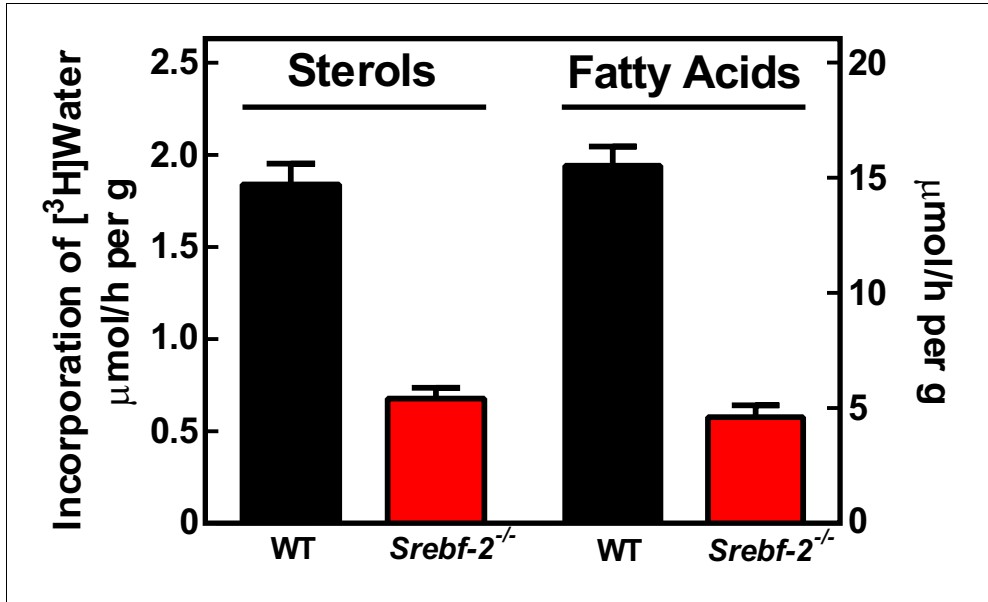

**Figure 3.** In vivo sterol and FA synthesis rates in livers of WT and hepatocyte-*Srebf-2*[-/-] mice. Six 4-month-old male WT and hepatocyte-*Srebf-2*[-/-] mice were injected intraperitoneally with 50 mCi [3]H-labeled water and rates of hepatic sterol and FA synthesis were determined as described in 'Materials and methods'.

missing SREBP-2-dependent endogenous LXR ligand is not one previously identified (*Supplementary file 1*).

In addition to LXR, *Srebf-1c* is transcriptionally activated by insulin, which is stimulated by feeding mice a high carbohydrate diet (*Horton et al., 1998*; *Shimomura et al., 1999*). To determine whether insulin-mediated activation of SREBP-1c was intact in hepatocyte-*Srebf-2*[-/-] livers, we subjected mice to a fasting/refeeding protocol using a high carbohydrate/low fat diet previously shown to induce SREBP-1c expression (*Horton et al., 1998*) (*Table 2*). In the fasted state, SREBP-1c levels are extremely low. As shown in *Table 3*, refeeding the high carbohydrate diet to fasted WT mice increased the expression of SREBP-1c mRNA in WT mice by 41-fold. In contrast, the SREBP-1c mRNA levels in livers from refed hepatocyte-*Srebf-2*[-/-] mice only increased to a level that was slightly higher than fasted WT mice. There were also blunted increases in the expression of FA synthesis genes in hepatocyte-*Srebf-2*[-/-] livers. The increase in FA synthesis mRNA expression that remained was likely mediated by ChREBP, a glucose-responsive transcription factor that can independently activate these genes (*Ishii et al., 2004*). These studies confirm that insulin-mediated induction of SREBP-1c requires intact LXR activity.

Deletion of SREBP-2 in the liver reduced the amount of LDLR mRNA by ~20% but there was an accompanying ~80% reduction in the mRNA level of PCSK9 (*Figure 2A*). PCSK9 is a secreted protein that degrades LDLRs in liver (*Lagace et al., 2006*). In livers of hepatocyte-*Srebf-2*[-/-] mice, the reduction in LDLR production was balanced by the reduction in PCSK9-mediated LDLR destruction, which ultimately led to no measurable change in steady-state LDLR protein levels (data not shown). Nevertheless, plasma cholesterol levels in hepatocyte-*Srebf-2*[-/-] mice were still 50% lower than those measured in WT mice (*Table 1*). To determine whether lower plasma cholesterol levels were a result of increased clearance of apoB-containing lipoproteins, we measured the [125]I-labeled LDL clearance. LDL was isolated from LDL receptor knockout mice and labeled the apoB with [125]I (*Horton et al., 1999*). As shown in *Figure 6A*, the clearance of LDL was identical in WT and hepatocyte-*Srebf-2*[-/-] mice. Therefore, the lower plasma and TG concentrations were likely a result of reduced VLDL production; therefore we measured rates of TG secretion in mice following the administration of Triton. As shown in *Figure 6B and C*, TG secretion rates from livers of hepatocyte-*Srebf-2*[-/-] mice reduced by 29%.

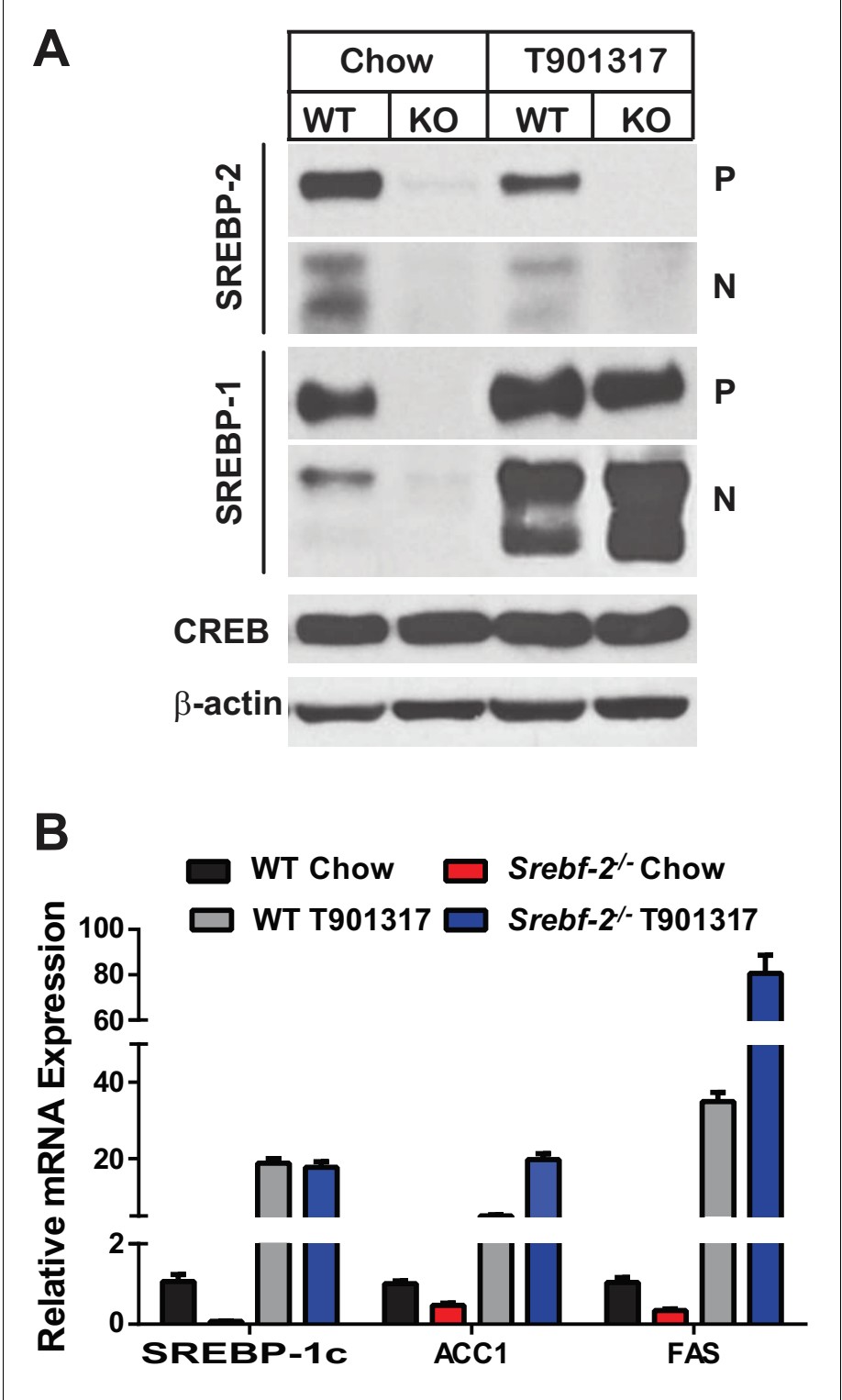

**Figure 4.** Levels of mRNAs and proteins in the livers of WT and hepatocyte-*Srebf-2*-/-mice fed chow diet supplemented with an LXR agonist. Mice 7–11 weeks of age were fed *ad libitum* chow or chow supplemented with 25 mg/kg of a LXR agonist (T901317) for three weeks prior to study. (**A**) Liver membrane and nuclear extract protein was made from each mouse and equal aliquots were pooled (total, 30 µg) and subjected to SDS-PAGE and immunoblot analysis as described in 'Materials and methods.' The precursor and nuclear form of SREBPs are denoted as P and N, respectively. (**B**) Total RNA from each mouse liver was subjected to real-time RT-PCR as

*Figure 4 continued on next page*

*Figure 4 continued*

described in 'Materials and methods.' Apo B was used as the invariant control. Values represent the amount of mRNA relative to those in the WT mice, which are arbitrarily assigned a value of 1. The following figure supplements are available for **Figure 4**.

The following figure supplement is available for figure 4:

**Figure 4 —figure supplement 1.** Liver mRNAs levels of WT and hepatocyte-*Scap*-/- mice fed chow diet supplemented with an LXR agonist.

## Discussion

The current data confirm that SREBP-2 is the primary transcriptional regulator of cholesterol biosynthesis in vivo. Deletion of *Srebf-2* in hepatocytes reduced the expression of all cholesterol biosynthetic genes and rates of hepatic cholesterol synthesis. Despite the deficiency in liver cholesterol content, no apparent additional mechanisms are present in the livers capable of restoring cholesterol levels to normal in the absence of SREBP-2. The unexpected finding in *Srebf-2* knockout livers was the marked reduction of SREBP-1c expression and genes involved in FA synthesis. While this manuscript was in preparation, Vergnes *et al.* (*Vergnes et al., 2016*) also reported that SREBP-1c expression and its regulated genes were reduced in livers of SREBP-2 hypomorphic mice. Here, dietary supplementation of cholesterol or an LXR agonist restored SREBP-1c expression and the mRNAs encoding the FA biosynthetic enzymes, which suggests that the loss of flux through the cholesterol biosynthetic pathway results in the loss of an endogenous LXR ligand required for normal LXR activity.

The current studies also have identified the first molecular mechanism linking cholesterol and FA synthesis in liver. This link requires SREBP-2 expression and is apparently supplied by an intermediate or product of cholesterol biosynthesis that serves as a ligand for LXR, which is required for SREBP-1c expression. Coupling cholesterol and FA synthesis may be important for the efficient esterification of cholesterol since oleic acid is the preferred substrate for the cholesterol esterifying enzyme ACAT (*Yang et al., 1997*). Cholesterol may also be required for the normal formation of the VLDL particle lipid core and thus linking cholesterol and FA synthesis might be necessary for efficient VLDL production by the liver.

LXR can be activated by 20(S)-hydroxycholesterol, 22(R)-hydroxycholesterol, 24(S)-hydroxycholesterol, 24(S),25-epoxycholesterol, 25-hydroxycholesterol, and 27-hydroxycholesterol as well as the cholesterol biosynthetic intermediate desmosterol (*Huang, 2014*; *Yang et al., 2006*). Supplementation of a synthetic LXR ligand or feeding cholesterol were both capable of restoring the SREBP-1c expression indicating that LXR was present and that if a ligand was provided it was capable of normally activating *Srebf-1c* transcription in the SREBP-2 deficient mice. However, we were unable to identify any changes in concentrations of known LXR ligands hepatocyte-*Srebf-2*-/- liver, indicating that the missing ligand is unique and not previously identified. Further studies will be required to identify this endogenous ligand.

The phenotype that resulted from the deletion of *Srebf-2* in hepatocytes was nearly indistinguishable from mice that lack Scap in hepatocytes (*Matsuda et al., 2001*; *Moon et al., 2012*). The only molecular signature we found that differed between hepatocyte-*Srebf-2*-/- and hepatocyte-*Scap*-/- livers was the retained expression of SREBP-1a and ACC2 in the livers of hepatocyte-*Srebf-2*-/- mice. These studies confirm that SREBP-1a has only a minor role in regulating basal and stimulated cholesterol and fatty acid synthesis in the liver. The phenotypic similarities between hepatocyte-*Srebf-2*-/- and hepatocyte-*Scap*-/- mice also suggest that blocking SREBP-2 action would be effective in preventing the development of hepatic steatosis in mice with insulin resistance and/or diabetes. The markedly reduced expression of lipogenic genes in hepatocyte-*Srebf-2*-/- livers and the blunted response of these genes to refeeding a high carbohydrate diet suggest that blocking SREBP-2 action would also be effective in preventing the development of hepatic steatosis induced by hyperinsulinemia, similar to the results obtained in Scap deficient mice that also lack leptin (*Moon et al., 2012*).

Plasma cholesterol and TG concentrations were also significantly lower in hepatocyte-*Srebf-2*-/- mice. The LDLR protein level was not reduced in hepatocyte-*Srebf-2*-/- livers despite a 20% reduction

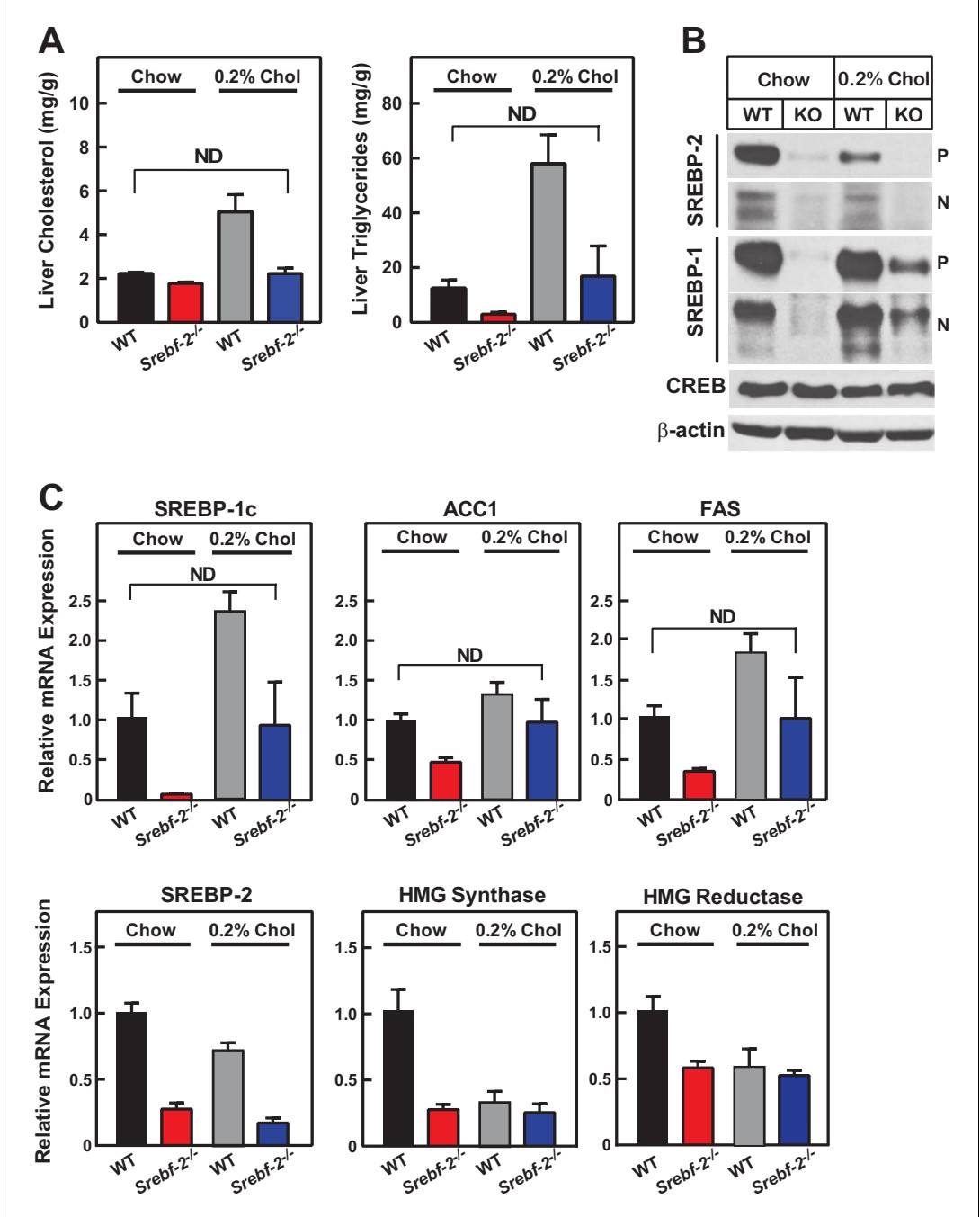

**Figure 5.** Liver lipid concentrations, mRNA, and protein levels in WT and hepatocyte-*Srebf-2⁻/⁻* mice fed chow or chow supplemented with cholesterol. Mice 7–11 weeks of age were fed chow (n = 6–7) or chow supplemented with 0.2% cholesterol (n = 6–7) for six weeks prior to study. (**A**) Liver cholesterol and TG concentrations were measured as described in 'Materials and methods.' (**B**) Equal aliquots of nuclear and membrane protein from each mouse liver were pooled (total, 30 μg) and subjected to SDS-PAGE and immunoblot analysis for the indicated protein as described in 'Materials and methods.' The precursor and nuclear form of SREBPs were denoted as P and N, respectively. (**C**) Total RNA from the livers of each mouse was subjected to real-time RT-PCR as described in 'Materials and methods.' Apo B was used as the invariant control. Values represent the amount of mRNA relative to those in WT mice, which are arbitrarily assigned a value of 1. * denotes a level of statistical significance of p<0.05 (Student's *t* test) between WT and hepatic-*Srebf-2⁻/⁻* mice, ND denotes no significant difference between the indicated groups.

**Table 2.** Phenotypic parameters in fasted and refed WT and hepatocyte-*Srebf-2*$^{-/-}$ mice. Male mice 9–12 wks of age were subjected to fasting and refeeding as described in 'Materials and methods.' Each value represents the mean ± SEM.

| Parameter | WT | | *Srebf-2*$^{-/-}$ | |
|---|---|---|---|---|
| | Fasted | Refed | Fasted | Refed |
| Number | 6 | 6 | 6 | 6 |
| Body weight (g) | 22.7 ± 1.4 | 25.8 ± 1.1 | 19.0 ± 1.3 | 21.7 ± 1.2* |
| Liver weight (g) | 0.92 ± 0.07 | 1.53 ± 0.19 | 0.82 ± 0.10 | 1.23 ± 0.13 |
| Liver cholesterol (mg/g) | 1.80 ± 0.08 | 1.02 ± 0.03 | 1.03 ± 0.05* | 0.71 ± 0.07* |
| Liver triglycerides (mg/g) | 52.6 ± 11 | 10.3 ± 1.8 | 33.2 ± 4.8 | 3.0 ± 0.5* |
| Plasma cholesterol (mg/dl) | 142 ± 9.0 | 90.2 ± 15 | 63.3 ± 7.3* | 43.6 ± 6.1* |
| Plasma triglyceride (mg/dl) | 142 ± 11 | 122 ± 16 | 58.5 ± 4.6* | 28.9 ± 3.8* |
| Plasma insulin (ng/ml) | 0.07 ± 0.01 | 1.00 ± 0.30 | 0.08 ± 0.02 | 0.48 ± 0.17 |
| Plasma glucose (mg/dl) | 184 ± 28 | 220 ± 14 | 121 ± 14 | 182 ± 16 |

* denotes a level of statistical significance of p<0.05 (Student's *t* test) between WT and hepatocyte-*Srebf-2* $^{-/-}$ mice.

in LDLR mRNA levels; however, the mRNA levels of PCSK9 were reduced by 80% in hepatocyte-*Srebf-2*$^{-/-}$ livers. Inasmuch as LDL clearance was not altered in hepatocyte-*Srebf-2*$^{-/-}$ mice, the reduced LDLR protein destruction by PCSK9 likely offset the reduced LDLR production since LDL clearance from the plasma of hepatocyte-*Srebf-2*$^{-/-}$ mice was not lower. Thus, the lower plasma lipid levels were a reflection of reduced VLDL secretion from the liver.

A potential additional benefit of inhibiting SREBP expression in the liver, independent of the reduction in hepatic TGs, is the reduced expression of *PNPLA3*. Polymorphisms in *PNPLA3* are associated with hepatic steatosis, nonalcoholic steatohepatitis, cirrhosis, and hepatocellular carcinoma in humans (*Romeo et al., 2008*; *Speliotes et al., 2010*). SREBP-1c is the only known transcriptional activator of PNPLA3 expression (*Huang et al., 2010*). In WT mice refed a high carbohydrate diet, PNPLA3 mRNA levels increased >200–fold above the fasted state, whereas in livers of hepatocyte-*Srebf-2*$^{-/-}$ mice PNPLA3 only increased 15–fold (*Table 3*). The blunted PNPLA3 expression in livers of hepatocyte-*Srebf-2*$^{-/-}$ mice was likely due to the accompanying loss of SREBP-1c since feeding the hepatocyte-*Srebf-2*$^{-/-}$ mice diets supplemented with an LXR agonist or cholesterol (data not shown) restored SREBP-1c and PNPLA3 expression. Studies by Hobbs and colleagues (*Li et al., 2012*; *Smagris et al., 2015*) previously demonstrated that high expression levels of the mutant PNPLA3 protein are required for the development of hepatic steatosis in mice. Thus, a reduction in mutant PNPLA3 expression as a result of inhibiting SREBP-2 or Scap may be of therapeutic benefit in individuals who carry the *PNPLA3* polymorphism.

The current report represents the last in a series of studies that we have carried out using genetically manipulated mice to elucidate the in vivo function of the SREBP family members (*Horton et al., 2002*). They confirm that SREBP-2 mediates the regulated expression of cholesterol biosynthetic genes and also controls steady-state tissue cholesterol concentrations by simultaneously regulating cholesterol synthesis and uptake from plasma and by modulating the expression of the LDLR and PCSK9. The physiological changes that result from deleting *Srebf-2* mirror that of mice that lack Scap in hepatocytes since we show that SREBP-2 expression is required to produce an LXR ligand required for normal SREBP-1c expression. The resulting phenotypes suggest that the inhibition of SREBP-2 or Scap in the liver, which reduces cholesterol and FA synthesis, may be therapeutically advantageous for the treatment of hypertriglyceridemia and nonalcoholic fatty liver disease.

**Table 3.** Gene expression in the livers of fasted and refed WT and hepatocyte-*Srebf-2*-/- mice. Total RNA from livers of each mouse liver described in **Table 2** was subjected to real-time RT-PCR as described in 'Materials and methods.' ApoB was used as the invariant control mRNA. Each value represents the amount of mRNA relative to that in fasted WT mice, which is arbitrarily defined as 1.

| | WT | | *Srebf-2*-/- | |
| --- | --- | --- | --- | --- |
| | Fasted | Refed | Fasted | Refed |
| *SREBP Pathway* | | | | |
| SREBP-2 | 1.0 ± 0.1 | 1.4 ± 0.1 | 0.1 ± 0.0 | 0.5 ± 0.1 |
| SREBP-1a | 1.0 ± 0.1 | 2.6 ± 0.3 | 1.2 ± 0.1 | 4.7 ± 1.2 |
| SREBP-1c | 1.0 ± 0.1 | 41 ± 2.0 | 0.2 ± 0.0 | 2.7 ± 1.5 |
| *Cholesterol Metabolism* | | | | |
| LDLR | 1.0 ± 0.0 | 3.0 ± 0.2 | 1.0 ± 0.1 | 2.3 ± 0.2 |
| HMG-CoA synthase | 1.0 ± 0.1 | 11 ± 1.8 | 0.7 ± 0.1 | 2.7 ± 0.8 |
| HMG-CoA reductase | 1.0 ± 0.0 | 11 ± 1.2 | 1.0 ± 0.1 | 4.1 ± 0.8 |
| Squalene synthase | 1.0 ± 0.1 | 4.3 ± 0.5 | 0.8 ± 0.1 | 1.1 ± 0.2 |
| *Fatty Acid Metabolism* | | | | |
| Acetyl-CoA Carboxylase1 | 1.0 ± 0.1 | 18 ± 2.3 | 0.7 ± 0.0 | 6.9 ± 1.4 |
| Fatty acid synthase | 1.0 ± 0.1 | 92 ± 7.6 | 0.4 ± 0.0 | 16 ± 6.0 |
| ELOVL6 | 1.0 ± 0.1 | 55 ± 7.4 | 0.7 ± 0.1 | 10 ± 2.8 |
| Stearoyl-CoA desaturase 1 | 1.1 ± 0.2 | 31 ± 5.4 | 0.0 ± 0.0 | 1.8 ± 1.0 |
| PNPLA3 | 1.3 ± 0.5 | 211 ± 43 | 1.9 ± 0.3 | 29 ± 7.8 |
| CHREBP | 1.0 ± 0.1 | 3.4 ± 0.2 | 0.7 ± 0.0 | 1.4 ± 0.2 |
| *Glucose Metabolism* | | | | |
| Glucokinase | 1.2 ± 0.3 | 51 ± 3.3 | 1.8 ± 0.3 | 17 ± 3.2 |
| G6PD | 1.0 ± 0.1 | 10 ± 2.1 | 2.6 ± 0.4 | 8.4 ± 3.2 |
| PEPCK | 1.0 ± 0.1 | 0.0 ± 0.0 | 1.1 ± 0.1 | 0.1 ± 0.0 |
| *Control* | | | | |
| ApoB | 1.0 ± 0.1 | 0.9 ± 0.0 | 1.0 ± 0.1 | 0.9 ± 0.1 |

# Materials and methods

## General supplies and measurements

Plasma concentrations of cholesterol, TGs, insulin, glucose, and free FAs, and liver cholesterol and TGs contents were measured as previously described (*Engelking et al., 2004*; *Ishibashi et al., 1993*; *Matsuda et al., 2001*). Liver sterol concentrations were determined using high performance liquid chromatography mass spectrometry (*McDonald et al., 2007*).

## Construction of a targeting vector for the conditional disruption of *Srebf-2*.

A conditional targeting vector of a replacement type was produced as follows. A *loxP* site was inserted into the promoter region of the *Srebf-2*~660 bp upstream of exon 1, and a *loxp; frt*-flanked *pgk-neo-pA* cassette was inserted into intron 1. Exon 1 encodes the first 29 amino acids of *Srebf-2*. The conditional targeting vector was constructed in five steps as follows: (1) A 1.1 kb fragment of intron 1 was generated by PCR from SM-1 ES cell genomic DNA with 5' primers that contained a HindIII sequence and a *loxP* site, and a 3' primer that contained a SalI sequence (5' primer, 5'-AAAAAAGCTTATAACTTCGTATAATGTATGCTATACGAAGTTATCCCGAAGCGGGGGCTGGGGGGCG TCGCGAG-3' and 3' primer, 5'-AAAAAGTCGACTTGTCACACTGTCTGGATGACCAAAATG-3'). This

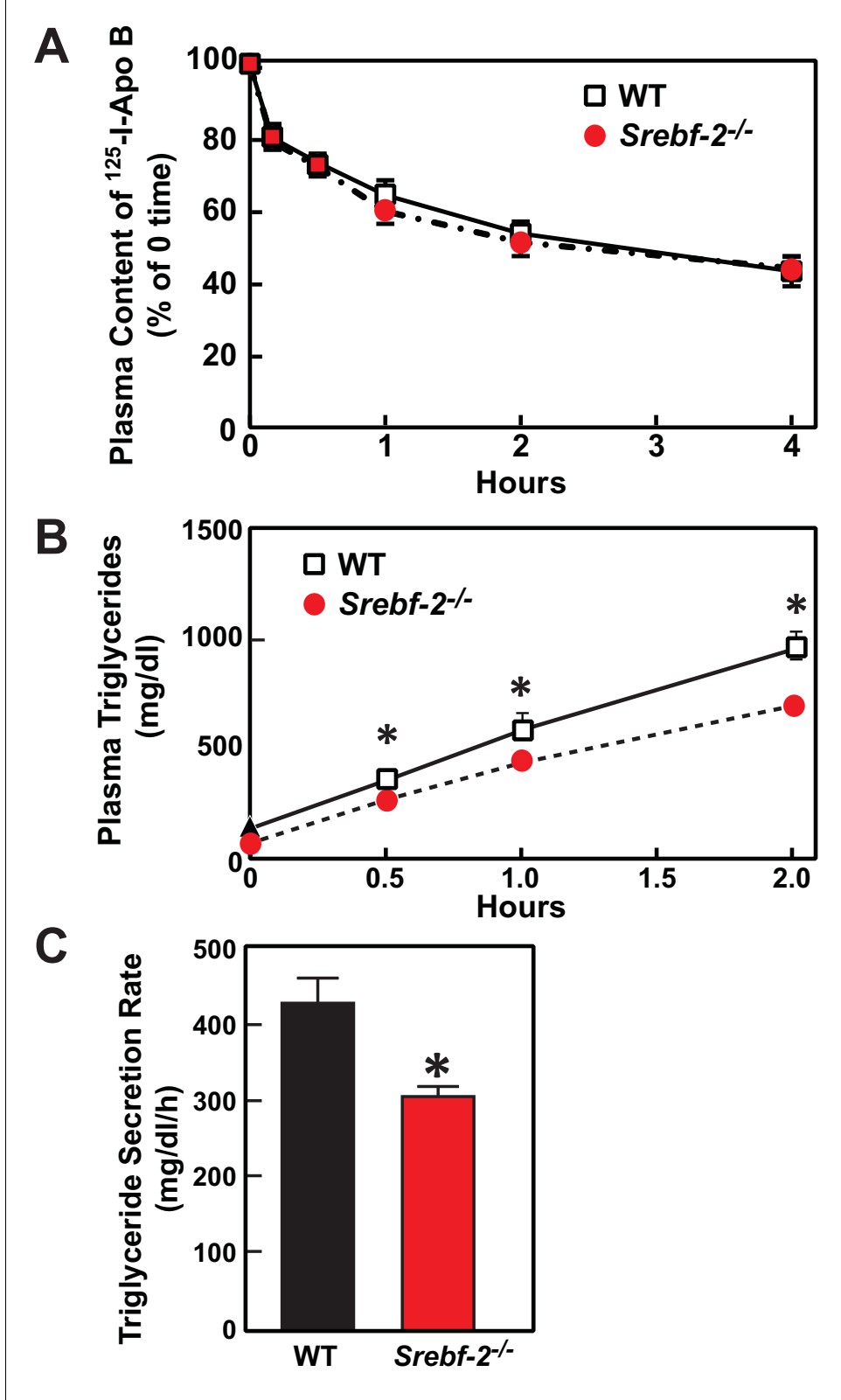

**Figure 6.** In vivo VLDL secretion and LDL clearance in WT and hepatocyte-*Srebf-2*[-/-] mice. (**A**) Eleven male mice (8 weeks of age) of each genotype were subjected to i.v. injection of [125]I-labeled LDL (15 μg of protein, 496 cpm/ng protein). Blood was obtained at 30 s (time 0) and 10, 30, 60, 120, and 240 min for the quantification of plasma content of [125]I-labeled apoB. Data were plotted as the percentage of 0 time value. (**B**) Five male mice (8 wks of *Figure 6 continued on next page*

*Figure 6 continued*

age) of each genotype were fasted for 4 hr prior to the study. Each mouse was injected i.v. with 10% triton-saline solution at 500 mg/kg. Plasma TG accumulation of each mouse at 0, 0.5, 1, and 2 hr after the triton injection were measured. (C) Plasma TG secretion rate during a detergent block of lipolysis was calculated for each mouse from the linear regression analysis of the time vs. TG concentration.

fragment was used as the short arm. (2) The HindIII- SalI fragment of the short arm and a BamHI-HindIII fragment containing a *loxP;frt* flanked *pgk-neo-pA* cassette excised from pGEMFRTNEO (provided by Joachim Herz, UT Southwestern) were ligated into the BamHI-SalI sites of pGEM-11Zf(+) (Promega, Madison, WI), yielding plasmid pBP2-KO1. (3) The middle arm contained a *loxP* site, and ~660 bp promoter, exon 1 and ~230 bp intron 1 of *Srebf-2*. It was generated by PCR using 5' primers that contained a NotI site and a *loxP* site, and 3' primer that contained a BamHI site (5' primer, 5'-AAAAAGCGGCCGCATAACTTCGTATAATGTATGCTATACGAAGTTATGATGCAG TGAGGTGACTGCAGGAGTGGG-3') and 3' primer, 5'-AAAAAGGATCCCCGCGGCGCCCACGAC TCCTCAG-3'). The NotI-BamHI fragment of the middle arm was ligated into the NotI and BamHI sites of pBP2-KO1, yielding plasmid pBP2-KO2. (4) Two copies of *hsv-tk* cassette were inserted into the SalI site of pBP2-KO2, yielding pBP2-KO3. (5) The long arm is a 7 kb fragment upstream of the promoter region of the *Srebf-2*. It was prepared by PCR using TaKaRa LA Taq DNA polymerase (Takara Shuzo, Shiga, Japan). The following NotI-containing PCR primers were used for amplification: 5' primer, 5'-AAAAAGCGGCCGCCTTGGTGAGGGCAGGCTGCAGGCCACTG-3' and 3' primer, 5'-AAAAAGCGGCCGCATCTTACAGGTAGTCGGTCACACTGCACAC-3'). The PCR fragment was digested with NotI and inserted into the NotI site of pBP2-KO3, resulting in the final *Srebf-2* conditional targeting vector, designated pBP2-KO4. The integrity of all plasmids was confirmed by restriction analysis and DNA sequencing.

## ES cell culture for the disruption of *Srebf-2*

Passage 8 SM-1 ES cells derived from 129S6/SvEv blastocysts were cultured on leukemia inhibitory factor-producing STO feeder cells (*Shimano et al., 1997*). On day 0, a total of $1 \times 10^7$ cells were transfected by electroporation (275 V, 330 µF, low resistance; GIBCO BRL Electroporator; Life Technologies, Gaithersburg, MD) with 50 µg of SfiI-linearized targeting vector and seeded onto γ-irradiated STO feeder cells. On day 2, ES cells were subjected to selection with 250 µg/ml of G418 (GIBCO BRL, NY). On day 4, ES cells were treated with 2.5 µM Ganciclovir (Bristol-Myers Squibb, Princeton, NJ) to select against random integration. G418 and Ganciclovir-resistant clones were isolated on day 10, and recombined clones were identified by PCR using P1 (5'-CCATCTTGTTCAA TGGCCGATCCCAT-3' from the 5' coding region of the *neo* gene) and P2 (5'-ACTTTAGCCAC TCCCACGTTCCAAGGAG-3' from the intron 1 of the *Srebf-2* gene outside of the targeting vector). The upstream *loxP* site in the promoter region was confirmed by PCR with primers P3 (5'-TGTACC TGATGCCTTACTGTGTTACTG-3' located ~900 bp upstream of exon 1 and P4 (5'-CTTAACAAGGTC TTGAGATCACCTGAG-3' located ~570 bp upstream of exon 1). The targeted clones were confirmed by Southern blot analysis using a 0.5 kb EcoRI-ApaI genomic DNA probe containing exon 1 and a 0.8 kb EcoRI-HindIII genomic DNA probe containing intron 1 sequence outside of the targeting vector.

## Generation of *Srebf-2^f/+* and *Srebf-2^f/f*; Albumin-Cre mice

One targeted ES clone containing a single *Srebf-2^flox/+* allele was injected into C57BL/6J blastocysts, yielding chimeric males whose coat color (*agouti*) indicated a contribution of ES cells from 50–100%. All six chimeric males with 75–95% were fertile, two of which produced offspring that carried the *Srebf-2^flox/+* allele through the germline. One line was established and used for further breeding. Mice carrying the floxed *Srebf-2* allele were genotyped by PCR using primers P3 and P4 (30 cycles, 94°C, 30 s; 60°C, 30 s; 72°C, 2 min). The WT allele produced a PCR product of 330 bp, and the floxed allele a product of 380 bp.

To generate tissue-specific *Srebf-2* deleted mice, mice heterozygous for the *Srebf-2^flox/+* allele (designated *Srebf-2^f/+*) were bred with Albumin-Cre transgenic mice to produce *Srebf-2^f/+*;albumin-Cre mice. The *Srebf-2^f/+*;albumin-Cre mice were bred with *Srebf-2^f/+* mice to generate *Srebf-2^f/f*;

albumin-Cre mice. The albumin-Cre transgene was identified by PCR using primers 5'-GGCCCACAC TGAAATGCTCAAATGGGAGAC-3' and 5'-GGTTACCCACTTCATTTTGCCAGAGGCTAG-3', which produces a 550 bp product. PCR conditions were the same as that for the genotyping of the floxed *Srebf-2* allele.

## Diet studies

All mice were housed in colony cages and maintained on a 12 hr light/12 hr dark cycle and fed Teklad Mouse/Rat Diet 2018 from Harlan Teklad Premier Laboratory Diets (Envigo, Madison, WI). For the cholesterol supplementation experiments, mice of each genotype were fed for six weeks with Teklad Mouse/Rat Diet 2018 supplemented with 0.2% cholesterol. For the LXR agonist (T901317) (Cayman Chemical, Ann Arbor, MI) administration studies, mice were fed *ad libitum* a powdered diet (Teklad Mouse/Rat Diet 2018) containing sufficient T901317 to provide a daily dose of ~25 mg/kg, assuming a 30 g mouse consumes 5 g of chow per day. In fasting refeeding studies, mice were subjected to a fasting and refeeding with a high carbohydrate/low fat diet as described (*Liang et al., 2002*). Specifically, one group of mice were fasted for 12 hr and a second group was fasted for 12 hr and then refed a high-carbohydrate/low-fat diet (TD 88122; Harlan Teklad) for 12 hr prior to study. The starting times for the feeding regimens were staggered so that all mice were sacrificed at the same time, which was at the end of the dark cycle.

## Quantitative real-time PCR

Total RNA was prepared from mouse livers with an RNA STAT-60 kit (Tel-Test, Friendswood, TX). cDNA was synthesized from 2 µg of DNase I-treated total RNA (DNA-free, DNA removal kit, Invitrogen, cat. no. 1906) using the Taqman reverse transcription reagents (Applied Biosystems, Carlsbad, CA) and random hexamer primers. Specific primers for each gene were designed by using PRIMER EXPRESS software (Applied Biosystems, Carlsbad, CA). The real-time RT-PCR contained, in a final volume of 20 µL, 20 ng of reverse-transcribed total RNA, 167 nM of the forward and reverse primers, and 10 µL of 2X SYBR Green PCR Master Mix Applied Biosystems, Carlsbad, CA). PCR reactions were carried out in 384-well plates using the ABI PRISM 7900HT Sequence Detection System (Applied Biosystems, Carlsbad, CA). All reactions were done in triplicate. The relative amount of all mRNAs was calculated using the comparative threshold cycle ($C_T$) method. Mouse apo B mRNA was used as the invariant control. The primers for real-time PCR were described previously (*Liang et al., 2002*; *Park et al., 2004*; *Yang et al., 2001*).

## Immunoblot analysis

Membrane and nuclear proteins were prepared from frozen livers as described (*Engelking et al., 2004*; *Moon et al., 2012*). Equal aliquots (8 µg) of protein from individual livers were pooled (total, 40 µg) and the proteins were subjected to SDS-PAGE on 8% gels and transferred to nitrocellulose membrane (Bio-Rad, Hercules, CA). Immunoblot analyses were performed using polyclonal anti-mouse SREBP-1 and the monoclonal anti-mouse SREBP-2 antibody as described (*Engelking et al., 2004*; *McFarlane et al., 2015*; *Moon et al., 2012*) using rabbit monoclonal anti-SREBP-1 (IgG-20B12) and anti-SREBP-2 (IgG-22D5) antibodies that were generated against bacterially produced, His-tagged proteins containing amino acids 32–250 of mouse SREBP-1a or SREBP-2. Antibody-bound bands were detected using the SuperSignal West Pico Chemiluminescent Substrate system (ThermoScientific, cat. no. 34080). Anti-mouse CREB (cAMP response element binding protein, Invitrogen) and anti-dog Calnexin (Enzo Life Science, Farmingdale, NY) antibodies were used as loading controls for nuclear and membrane proteins, respectively.

## In vivo hepatic lipid synthesis

Rates of sterol and FA synthesis in liver were determined using [3]H-labeled water as described (*Shimano et al., 1996*).

## In vivo VLDL secretion

Mice were fasted for 4 hr and injected with 10% Triton WR-1339/saline solution (Tyloxapol; Sigma-Aldrich) (500 mg/kg) via the retro-orbital vein. Blood was collected from the tail vein at 0, 0.5, 1, and

2 hr after the triton injection and assayed for plasma levels of TGs. The plasma TG secretion rate was calculated from the linear regression analysis of the time vs. TG concentration.

## Plasma clearance of $^{125}$I-LDL

Mouse LDL (density 1.019–1.063 g/ml) was obtained from pooled *Ldlr*$^{-/-}$ mouse plasma by sequential ultracentrifugation and radiolabeled with sodium $^{125}$I. Clearance of the labeled LDL from plasma was studied as previously described (*Horton et al., 1999*; *Rashid et al., 2005*). Briefly, recipient mice were anesthetized with sodium pentobarbital and received a bolus of 0.1 ml of saline containing 15 µg of $^{125}$I-LDL (496 cpm/ng protein; ~53% labeled on apo B) via the right jugular vein. Blood was collected from the left jugular vein at 0.5, 10, 30, 60, 120, and 240 min after the injection. Remaining plasma $^{125}$I-labeled apo B was determined by γ-scintillation spectrometry after isopropanol precipitation.

## Acknowledgements

We thank Tuyet Dang, Bonne Thompson, Marcus Thornton, and Judy Sanchez, for excellent technical assistance and Jian Yang for assistance in the production of the L-*Srebf-2*$^{-/-}$ mice. We also thank Drs. Joseph L Goldstein and Michael S Brown for helpful suggestions throughout the project. This work was supported by grants from the National Institutes of Health HL-20948.

## Additional information

### Funding

| Funder | Grant reference number | Author |
|---|---|---|
| National Institutes of Health | HL-20948 | Jay D Horton |

The funders had no role in study design, data collection and interpretation, or the decision to submit the work for publication.

### Author contributions

SRo, Conceptualization, Data curation, Formal analysis, Validation, Investigation, Visualization, Methodology, Writing—original draft, Writing—review and editing; VAC, Data curation, Formal analysis, Validation, Methodology; SRa, Data curation, Methodology, Writing—original draft; NNA, Data curation, Formal analysis, Supervision, Validation, Methodology; JGM, Data curation, Formal analysis, Methodology; GL, Data curation, Formal analysis, Supervision, Investigation, Methodology; Y-AM, Conceptualization, Resources, Data curation, Supervision, Validation, Investigation, Methodology, Writing—original draft; REH, Resources, Methodology; JDH, Conceptualization, Data curation, Formal analysis, Supervision, Funding acquisition, Investigation, Methodology, Project administration, Writing—review and editing

### Author ORCIDs

Shunxing Rong, http://orcid.org/0000-0002-4948-7222
Jay D Horton, http://orcid.org/0000-0002-7778-1074

### Ethics

Animal experimentation: All animal experiments were performed with approval of the Institutional Animal Care and Research Advisory Committee at UT Southwestern.

## Additional files

### Supplementary files

• Supplementary file 1. Liver sterol and oxysterol concentrations. Male mice (12–13 wks old in study 1 and 16–18 wks old in study 2) fed chow *ad lib* were sacrificed and liver sterols and oxysterols were

measured with mass spectrometry. Each value represents mean ± SEM. * denotes the level of statistical significance of p<0.05 (Student's *t* test) between WT and hepatocyte-*Srebf-2*<sup>-/-</sup> mice.

• Supplementary file 2. Relative mRNA expression in livers of WT and hepatocyte-*Srebf-2*<sup>-/-</sup> mice of *Table 1*. Liver was snap frozen and mRNA prepared from the mice shown in *Table 1*. Each value represents mean ± SEM. * denotes the level of statistical significance of p<0.05 (Student's *t* test) between WT and hepatocyte-*Srebf-2*<sup>-/-</sup> mice.

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
