## [Decision Letter]

Thank you for submitting your article "Hepatic expression of SREBP-1c requires SREBP-2-mediated generation of a sterol ligand for LXR" for consideration by *eLife*. Your article has been favorably evaluated by Harry Dietz (Senior Editor) and three reviewers, one of whom, Peter Tontonoz, is a member of our Board of Reviewing Editors. The reviewers have opted to remain anonymous.

The reviewers have discussed the reviews with one another and the Reviewing Editor has drafted this decision to help you prepare a revised submission.

Summary:

This manuscript describes unexpected alteration of the SREBP-1 pathway in SREBP-2 liver specific knockout mice. The authors found that elimination of SREBP-2 in liver significantly reduced the activity of the SREBP-1c pathway and the expression of genes involved in fatty acid synthesis regulated by SREBP- 1c. Feeding mice with either LXR ligand or high cholesterol diet relieved SREBP-1c suppression. These data suggested that deletion of SREBP-2 reduced the production of an endogenous sterol ligand required for LXR activity and SREBP-1c expression. A puzzling aspect is that putative in vivo LXR oxysterol ligands are not reduced in the SREBP-2 KO livers and some oxysterol species are even higher in the KO. In general, the data presented in this manuscript is clear, and is consistent with the hypothesis put forward by the authors.

The reviewers were in agreement that the work was interesting and potentially of interest to the general audience of *eLife*. The studies were judged to be technically sound and the reviewers believed that the majority of the conclusions drawn were supported by the data. At the same time, the review process identified several opportunities to strengthen the work.

Essential revisions:

1) In this manuscript, the comparison of phenotypes was performed between wild type and Srebf-2 fl/fl, Alb-Cre mice. A more appropriate control group should be the Srebf-2 fl/fl mice, instead of the wild type mice. The authors should address if there is any difference in the concerned phenotypes between the wild type mice and the Srebf-2 fl/fl mice.

2) Results, last paragraph and Figure 6: It is surprising that the authors sought to explain a 50% reduction in plasma cholesterol in the SREBP2 KO mice on the basis of a reduction in LDL; LDL levels in mice are not high enough to explain this level of reduction. Wouldn't it be more likely that it is due to a reduction in ABCA1 expression and thus a loss of HDL? Does T901317 restore serum cholesterol levels?

3) As useful information for the research community, it would be helpful if the authors could discuss the phenotype of heterozygous liver specific Srebf-2 knockout mice.

---

## [Author Response]

*Essential revisions:*

*1) In this manuscript, the comparison of phenotypes was performed between wild type and Srebf-2 fl/fl, Alb-Cre mice. A more appropriate control group should be the Srebf-2 fl/fl mice, instead of the wild type mice. The authors should address if there is any difference in the concerned phenotypes between the wild type mice and the Srebf-2 fl/fl mice.*

In all studies, offspring of Srebp-2 ^f/f^ mice crossed with Srebp-2 ^f/f^, Alb-Cre transgenic mice were used. The Srebp-2 ^f/f^ is what we referred to as wild type controls and Srebp-2 ^f/f^; Alb-Cre transgenic mice as liver specific SREBP-2 knockout mice (L-SREBP-2^-/-^). We did not make this clear in the manuscript. This information in added the revised version the first paragraph of the Results.

*2) Results, last paragraph and Figure 6: It is surprising that the authors sought to explain a 50% reduction in plasma cholesterol in the SREBP2 KO mice on the basis of a reduction in LDL; LDL levels in mice are not high enough to explain this level of reduction. Wouldn't it be more likely that it is due to a reduction in ABCA1 expression and thus a loss of HDL? Does T901317 restore serum cholesterol levels?*

We agree with the reviewer that the reduction of plasma cholesterol levels in this study was due to the reduced HDL-C rather than the LDL. The purpose of the LDL clearance study (Figure 6), was simply to demonstrate that there was no increased clearance of LDL-C since the deletion of SREBP-2 effected the expression of the LDLR and PSCSK9. The fact that there was no difference supports the results of Figure 6 that showed there was a marked reduction in VLDL production and we feel this is the most likely cause of the lower plasma lipid levels. We did measure the expression of ABCA1 but found no changes. The LXR agonist does restored serum cholesterol levels of L-SREBP-2^-/-^ mice.

*3) As useful information for the research community, it would be helpful if the authors could discuss the phenotype of heterozygous liver specific Srebf-2 knockout mice.*

We have added a statement regarding the lack of phenotype in SREBP-12 heterozygous mice in the Introduction.